# Content-Aware SLIC Super-Pixels for Semi-Dark Images (SLIC++)

**DOI:** 10.3390/s22030906

**Published:** 2022-01-25

**Authors:** Manzoor Ahmed Hashmani, Mehak Maqbool Memon, Kamran Raza, Syed Hasan Adil, Syed Sajjad Rizvi, Muhammad Umair

**Affiliations:** 1High Performance Cloud Computing Center (HPC3), Department of Computer and Information Sciences, Universiti Teknologi PETRONAS, Seri Iskandar 32610, Malaysia; manzoor.hashmani@utp.edu.my (M.A.H.); muhammad_17008606@utp.edu.my (M.U.); 2Faculty of Engineering Science and Technology, Iqra University, Karachi 75600, Pakistan; kraza@iqra.edu.pk (K.R.); hasan.adil@iqra.edu.pk (S.H.A.); 3Department of Computer Science, Shaheed Zulfiqar Ali Bhutto Institute of Science and Technology, Karachi 75600, Pakistan; sshussainr@gmail.com

**Keywords:** clustering, similarity measure, geodesic measure, Euclidean measure

## Abstract

Super-pixels represent perceptually similar visual feature vectors of the image. Super-pixels are the meaningful group of pixels of the image, bunched together based on the color and proximity of singular pixel. Computation of super-pixels is highly affected in terms of accuracy if the image has high pixel intensities, i.e., a semi-dark image is observed. For computation of super-pixels, a widely used method is SLIC (Simple Linear Iterative Clustering), due to its simplistic approach. The SLIC is considerably faster than other state-of-the-art methods. However, it lacks in functionality to retain the content-aware information of the image due to constrained underlying clustering technique. Moreover, the efficiency of SLIC on semi-dark images is lower than bright images. We extend the functionality of SLIC to several computational distance measures to identify potential substitutes resulting in regular and accurate image segments. We propose a novel SLIC extension, namely, SLIC++ based on hybrid distance measure to retain content-aware information (lacking in SLIC). This makes SLIC++ more efficient than SLIC. The proposed SLIC++ does not only hold efficiency for normal images but also for semi-dark images. The hybrid content-aware distance measure effectively integrates the Euclidean super-pixel calculation features with Geodesic distance calculations to retain the angular movements of the components present in the visual image exclusively targeting semi-dark images. The proposed method is quantitively and qualitatively analyzed using the Berkeley dataset. We not only visually illustrate the benchmarking results, but also report on the associated accuracies against the ground-truth image segments in terms of boundary precision. SLIC++ attains high accuracy and creates content-aware super-pixels even if the images are semi-dark in nature. Our findings show that SLIC++ achieves precision of 39.7%, outperforming the precision of SLIC by a substantial margin of up to 8.1%.

## 1. Introduction

Image segmentation has potential to reduce the image complexities associated with processing of singular image primitives. Low-level segmentation of an image in non-overlapping set of regions called super-pixels helps in pre-processing and speeding up further high-level computational tasks related to visual images. The coherence feature of super-pixels allows faster architectural functionalities of many visual applications including object localization [1], tracking [2], posture estimation [3], recognition [4,5], semantic segmentation [6], instance segmentation [7], and segmentation of medical imagery [8,9]. These applications will be aided by super-pixels in terms of boosted performances, as the super-pixels put forward only the discriminating visual information [10].

Low-level segmentation tends to result in incorrect segmentations if the visual image has high pixel intensities; these high pixeled values are usually the biproduct of visual scenes captured in low lighting conditions, i.e., semi-dark images and dark images. The obtained incorrect super-pixels are attributed to the underlying approach used for the final segment creation. The existing super-pixel methods fail due to incorrect pixel manipulations for base operational functionality. The currently employed pixel manipulation relies on straight line differences for super-pixel creation. These straight-line difference manipulations fail to retain the content-aware information of the image. The results are further degraded if the image has low contrasted values, which result in no clear discrimination among the objects present. The existing super-pixel creation methods are divided into two categories based on the implemented workflow. The two categories are graph-based and gradient ascent based [10,11]. The former, focuses on minimization of cost function by grouping and treating each pixel of the image as a graph node. The later, iteratively processes each image pixel using clustering techniques until convergence [12]. One of the typical features observed for identification of super-pixel accuracy is regularity, i.e., to what extent the super-pixel is close to the actual object boundary of the image. All the graph-based methods for super-pixel calculations suffer from poorly adhered super-pixels which result in irregularity of segments presented by super-pixels [11]. Additionally, graph-based methods are constrained by the excessive computational complexity and other initialization parameters. Whereas, the gradient-ascent methods are simplistic in nature and are recommended in the literature due to resultant high performance and accuracy [13,14]. However, there are some issues associated with content irrelevant manipulation of singular pixels to form resultant super-pixels. Some of the key features of using super-pixel segmentations are:Super-pixels abstraction potentially decreases the overhead of processing each pixel at a time.Gray-level or color images can be processed by a single algorithm implementation.Integrated user control provided for performance tuning.

With these advantages of super-pixels, they are highly preferred. However, super-pixel abstraction methods backed by gradient-ascent workflows are also limited in their working functionality to retain the contextual information of the given image [15]. The contextual information retainment is required to achieve the richer details of the visual image. This loss of contextual information is caused by flawed pixel clusters created based on Euclidean distance measure [16]. As the Euclidean distance measures calculates straight line differences among pixels which ends up in irregular and lousy super-pixels. Moreover, further degradation can be expected to process the semi-dark images where high pixel intensities along with no clear boundaries are observed. In such scenarios, the propagation of inaccurate super-pixels will affect the overall functionality of automated solutions [11]. To overcome this problem of information loss for creation of compact and uniform super-pixels, we propose content-aware distance measures for image pixel cluster creation. The content-aware distance measure as the core foundational component of gradient-ascent methods for super-pixel creation will not only help in alleviation of information loss, but it will also help in preserving less observant/perceptually visible information of semi-dark images. The state-of-the-art methods for super-pixels creation have not been analyzed exclusively on the semi-dark images which further raises the concerns related to segmentation accuracy. In nutshell, the problems in existing segmentation algorithms are:Absence of classification of state-of-the-art methods based on low level manipulations.Inherited discontinuity in super-pixel segmentation due to inconsistent manipulations.Unknown pixel grouping criteria in terms of distance measure to retain fine grained details.Unknown effect of semi-dark images on final super-pixel segmentations.

To resolve these issues, the presented research exclusively presents multifaceted study offering following features:Classification of Literary Studies w.r.t Singular Pixel Manipulation Strategies:

The categorization of the existing studies is based on entire image taken as one entity. The image entity can represent either graph or a feature space to be clustered, i.e., graph-based or gradient-ascent based methodology for pixel grouping. To the best of our knowledge, there has been no study that categorizes existing studies based on the manipulation strategy performed over each pixel. We present the detailed comparative analysis of existing research highlighting their core functionality as the basis for classification.

2.Investigation of Appropriate Pixel Grouping Scheme:

The grouping scheme backing the image segmentation module being crucial component can highly affect the accuracy of the entire model. For this reason, to propose a novel extension as a generalized solution of all types of images including semi-dark images, we present a detailed qualitative investigation of up to seven distance measures for grouping pixels to create super-pixels. The investigation resulted in shortlisted pixel grouping measures to retain fine grained details of the visual image.

3.Novel Hybrid Content-Aware Extension of SLIC—SLIC++:

SLIC, being the simplest and fastest solution for the pixel grouping, remains the inspiration, and we enhance the performance of SLIC by adding content-aware feature in its discourse. The proposed extension holds the fundamental functionality with improved features to preserve content-aware information. The enhancement results in better segmentation accuracy by extracting regular and continuous super-pixels for all scenarios including semi-dark scenarios.

4.Comprehensive Perceptual Results focusing Semi-dark Images:

To assess the performance of the proposed extension SLIC++ for extraction of the richer information of the visual scene, we conduct experiments over semi-dark images. The experimental analysis is benchmarked against the standard super-pixel creation methods to verify that incorporating content-aware hybrid distance measure leads to improved performance. The perceptual results further conform better performance, the scalability, and generalizability of results produced by SLIC++.

### Paper Organization

The remainder of paper is organized as follows: Section 2 presents the prior super-pixel creation research and its relevance to semi-dark technology. In this section we also present critical analysis of studies proposed over period of two decades and their possible applicability for semi-dark imagery. We also critically analyze two closely related studies and highlight the difference among them w.r.t SLIC++. Section 3 describes the extension hypothesis and the final detailed proposal for super-pixel segmentation of semi-dark images. Section 4 presents the detailed quantitative and qualitative analysis of SLIC extension against state-of-the-art algorithms validating the proposal. Section 5 discusses the applicability of the proposed algorithm in the domain of computer vision. Finally, Section 6 concludes the presented research and points out some future directions of research.

## 2. Literature Review

### 2.1. Limited Semi-Dark Image Centric Research Focusing Gradient-Ascent Methods

The gradient ascent methods are also called clustering-based methods. These methods take the input image and rasterize the image. Then, based on the local image cues such as color and spatial information the pixels are clustered iteratively. After each iteration, gradients are calculated to refine the new clusters from the previously created clusters [14]. The iterative process continues till the algorithm converges after the gradients stop changing, thus named gradient-ascent methods. A lot of research has been already done in the domain of gradient-ascent methods; the list of these methods is presented in Table 1.

The gradient ascent methods for super-pixel creation seems to be a promising solution due to their simplicity of implementation, speed of processing and easy adaptation for handling the latest demands of complex visual image scenarios. However, the concerns associated with underlying proper extraction strategies is one of the challenging aspects to cater the dynamic featural requirements imposed by complex visual image scenarios such as semi-dark images.

### 2.2. Critical Analysis of Gradient-Ascent Super-Pixel Creation Algorithms Based on Manipulation Strategy

For the critical analysis we have considered gradient ascent based super-pixel algorithms presented over period of two decades ranging from 2001 through 2021. The studies are retrieved from Google Scholar’s repository with keywords including super-pixel segmentation, pixel abstraction, content sensitive super-pixel creation, content-aware super-pixel segmentation. The search resulted in a lot of segmentation related studies in domain of image processing including the basic image transformations along with related super-pixel segmentation studies. For the critical analysis, the studies mentioning clustering based super-pixel creations were shortlisted due to their relevance with proposed algorithm. The key features of these studies are critically analyzed and comprehensively presented in Table 1, along with the critiques for respective handling concerns associated with semi-dark imagery.

#### Key Takeaways

The critical analysis presented in Table 1 uncovers the fact that recent research explicitly points towards the need of segmentation algorithm which considers the content relevant super-pixel segmentations. To accomplish this task several techniques are proposed with incorporation of prior transformations of image via deep learning methods, simple image processing, and probabilistic methods. Mostly the research uses and conforms the achievements of Simple Linear Iterative Clustering (SLIC). Furthermore, most of the research studies are using SLIC algorithm for super-pixel creation as base mechanisms with added features. Generally, the algorithms proposed in last decade have computational complexity of $O\left(N\right)$, whereas if neural networks are employed for automation of required parameter initialization, then the complexity becomes $O\left(N^{No.\;of\;layers}\right)$. All the proposed algorithms use two distance measures for final super-pixel creation, i.e., Euclidean, or geodesic distance measure. However, all these studies have not mentioned the occurrence of semi-dark images and their impact on the overall performance. It is estimated that huge margin of the existing image dataset already includes the problem centric image data. The Berkeley dataset that is substantially used for performance analysis of super-pixel algorithms contains up to 63% semi-dark images. The proposed study uses the semi-dark images of Berkeley dataset for benchmarking analysis of the SLIC++ algorithm.

### 2.3. Exclusiveness of SLIC++ w.r.t Recent Developments

The recent studies substantially focus on super-pixels with the induced key features of content sensitivity and adherence of the final segmentations; consequently several related research studies have been proposed. Generally, the desired features are good boundary adherence, compact super-pixel size and low complexity. The same features are required for super-pixel segmentation of semi-dark images. In this section we briefly review the recent developments which are closely related to our proposed method for creating super-pixel in semi-dark images.

BASS (Boundary-Aware Super-pixel Segmentation) [28], is closely related to the methodology that we have chosen, i.e., incorporation of content relevant information in the final pixel labeling which ends up with the creation of super-pixels. However, the major difference resides in the initialization of the super-pixel seeds/centers. BASS recommends the usage of forest classification method prior to the super-pixel creation. This forest classification of image space results with the creation of a binary image with highlighted boundary information over the image space. This boundary information is then utilized to aid the initialization process of the seed/ cluster centers. Theoretically, the problem with this entire configuration is additional complexity of boundary map creation which raise the complexity from $O\left(N\right)$ to $O\left(NlogN\right)$. This boundary map creation and its associated condition of addition and deletion of seeds is expected to further introduce undesired super-pixel feature under-segmentation. The under-segmentation might take place due to easy seed deletion condition and difficult addition condition which means more seeds would be deleted and less would be added. This aspect of the study is not desired for the super-pixel creation in semi-dark scenarios. On the contrary we propose regularly distributed seeds along with usage of both the recommended distance measures without prior image transformation which further reduces the overall complexity. Finally, we also propose the usage of geodesic distance for color components of the pixel rather than only using it for spatial component.

Intrinsic manifold SLIC [30], is an extension of manifold SLIC which proposes usage of manifolds to map high dimensional image data on the manifolds resembling Euclidean space near each point. IMSLIC uses Geodesic Centroidal Voronoi Tessellations (GCVT) this allows flexibility of skipping the post-processing heuristics. For computation of geodesic distance on image manifold weighted image graph is overlayed with the same graph theory of edges, nodes, and weights. For this mapping 8-connected neighbors of pixel are considered. This entire process of mapping and calculation of geodesic distances seems complex. The theoretical complexity is $O\left(N\right)$, however with the incorporation of image graph the computational complexity will increase. Moreover, the conducted study computes only geodesic distance between the pixels leaving behind the Euclidean counterpart. With substantially less complexity we propose to implement both the distance measures for all the crucial pixel components.

### 2.4. Summary and Critiques

The comprehensive literature survey is conducted to benefit the readers and provide a kickstart review of advancements of super-pixel segmentation over the period of two decades. Moreover, the survey resulted in critical analysis of existing segmentation techniques which steered the attention to studies conducted for adverse image scenarios such as semi-dark images. Arguably, with the increased automated solutions the incoming image data will be of dynamic nature (including lighting conditions). To deal with this dynamic image data, there is a critical need of a super-pixel segmentation technique that takes into account the aspect of semi-dark imagery and results in regular and content-aware super-pixel segmentation in semi-dark scenarios. The super-pixel segmentation techniques currently employed for the task suffer from two major issues, i.e., high complexity, and information loss. The information loss associated with the gradient-ascent methods is attributed to restrictions imposed due to usage of Euclidean image space which totally loses the context of the information present in the image by calculating straight line differences. Many attempts have been made to incorporate CNN probabilistic methods in super-pixel creation methods to optimize and aid the final segmentation results. However, to the best of our knowledge there has been no method proposed exclusively for semi-dark images scenarios keeping the simplicity and optimal performance intact.

In following sections, we describe the preliminaries which are the base for the proposed extension of SLIC namely SLIC++. We also present several distance measures incorporated in base SLIC algorithm namely SLIC+ to analyze the performance for semi-dark images.

## 3. Materials and Methods

### 3.1. The Semi-Dark Dataset

For the analysis of the content-aware super-pixel segmentation algorithm, we have used the state-of-the-art dataset which has been used in the literature for years now. The Berkeley image dataset [39] has been used for the comprehensive analysis and benchmarking of the proposed SLIC++ algorithm with the state-of-the-art algorithms. The Berkeley image dataset namely BSDS 500 has got five hundred images overall, whereas the problem under consideration is of semi-dark images. For this purpose, we have initially extracted semi-dark images using RPLC (Relative Perceived Luminance Classification) algorithm. The labels are created based on the manipulation of color model information, i.e., Hue, Saturation, Lightness (HSL) [40]. The final semi-dark images extracted from the BSDS-500 dataset turned out to be 316 images. Each image has resolution of either 321 × 481 or 481 × 321 dimensions. The BSDS-500 image dataset provides the basis for empirical analysis of segmentation algorithms. For the performance analysis and boundary detection, the BSDS-500 dataset provides ground-truth labels by at least five human annotators on average. This raises questions about the selection of annotation provided by the subjects. To deal with this problem, we have performed a simple logic over the image ground truth labels. All the image labels are iterated with ‘OR’ operation to generate singular ground truth image label. The ‘OR’ operation is performed to make sure that the final ground truth is suggested by most of the human annotators. Finally, every image is segmented and benchmarked against this single ground truth labeled image.

### 3.2. Desiderata of Accurate Super-Pixels

Generally, for super-pixel algorithms there are no definite features for super-pixels to be accurate. The literary studies refer accurate super-pixels in terms of boundary adherence, connectivity of super-pixels, super-pixel partitioning, compactness, regularity, efficiency, controllable number of super-pixels and so on [13,41,42]. As the proposed study is research focused on semi-dark images, we take into account features that are desired for conformation of accurate boundary extraction in semi-dark images.

**1.** 
**Boundary Adherence**


The boundary adherence is the measure to compute the accuracy to which the boundary has been extracted by super-pixels against boundary image or ground-truth images. The idea is to preserve information as much as possible by creating super-pixels over the image. The boundary adherence feature is basically a measure that results how accurately the super-pixels have followed the ground-truth boundaries. This can be easily calculated by segmentation quality metrics precision-recall.

**2.** 
**Efficiency with Less Complexity**


As super-pixel segmentation algorithms are now widely used as preprocessing step for further visually intelligent tasks. The second desired feature is efficiency with less complexity. The focus should be creation of memory-efficient and optimal usage of processing resources so that more memory and computational resources can be used by subsequent process. We take into account this feature and propose an algorithm that uses exactly same resources as of Basic SLIC with added distance measures in its discourse.

**3.** 
**Controllable Number of Super-Pixels**


The controllable number is super-pixels is a desired feature to ensure the optimal boundary is extracted using the computational resources ideally. The super-pixel algorithms are susceptible to this feature that is number of super-pixels. The number of super-pixels to be created can directly impact the overall algorithm performance. The performance is degraded in terms of under-segmentation or over-segmentation error. In the former one, the respective algorithm fails to retrieve most of the boundaries due to the smaller number of super-pixels to be created, whereas the latter one retrieves maximum boundary portions of the ground-truth images but there is surplus of computational resources.

Nevertheless, as mentioned earlier there is a huge list of accuracy measures and all those measures refer to different segmentation aspects and features. The required features and subsequent accuracy measures to be reported depend on the application of algorithm. For semi-dark image segmentation, it is mandatory to ensure that most of the optimal boundary is extracted and this requirement can be related to precision-recall metrics.

### 3.3. SLIC Preliminaries

Before presenting the SLIC++, we first introduce base functionality of SLIC. The overall functionality is based on creation of restricted windows in which the user defined seeds are placed, and clustering of image point is performed in this restricted window. This restricted window is called Voronoi Tessellations [43]. Voronoi Tessellations is all about partitioning the image plane into convex polygons. This polygon is square in case of SLIC initialization windows. The Voronoi tessellations are made such that each partition has one generating point and all the point in the partition are close to the generating point or the mass center of that partition. As the generating point lies in the center these partitions are also called Centroidal Voronoi Tessellation (CVT). The SLIC algorithm considers CIELAB color space for the processing, where every pixel p on image I is presented by color components and spatial components as cp=lp, ap, bp being colour components and pu,v being spatial components. For any two pixels SLIC measures straight line difference or Euclidean distance between the two pixels for the entire image space ℝ5.

The spatial distance between two pixels is given by $d_{s}$ and color component distance $d_{c}$ are given in Equations (1) and (2).
(1)$$d_{s}=\sqrt{{\left(u_{1}-u_{2}\right)}^{2}+{\left(v_{1}-v_{2}\right)}^{2}}$$

And,
(2)$$d_{c}=\sqrt{{\left(l_{1}-l_{2}\right)}^{2}+{\left(a_{1}-a_{2}\right)}^{2}+{\left(b_{1}-b_{2}\right)}^{2}}$$

Here $d_{s}$ and $d_{c}$ represent Euclidean distance between pixel $p_{1}$ and $p_{2}$. Instead of simple Euclidean, SLIC uses distance term infused with Euclidean norm given by Equation (3).
(3)$$D_{s}=d_{c}+\frac{m}{S}d_{s}$$

The final distance term is normalized using interval S and m provides control over the super-pixel compactness which results in perceptually meaningful distance with balanced aspect of spatial and color components. Provided the number of super-pixels K seeds ${\left(s_{i}\right)}_{i=1}^{K}$ are evenly distributed in over the image I clusters are created in restricted regions of Voronoi Tessellations. The initialization seed are placed in image space within a window of 2S×2S having center $s_{i}$. After that simple K-means is performed over the pixels residing in the window to its center. SLIC computes the distance between pixels using Equation (3) and iteratively processes the pixels until convergence.

### 3.4. The Extension Hypothesis—Fusion Similarity Measure

The super-pixels created by the SLIC algorithm basically uses the Euclidean distance measure to create pixel clusters or the super-pixels based on the seed or cluster centers. The Euclidean distance measure takes into account the similarity among pixels using straight line differences among cluster centers and the image pixels. This property of distance measure results in distortion of extracted boundaries of image. The reason is measure remains same no matter if there is a path along the pixels. The path along the pixels will result in smoother and content relevant pixels [16,36]. The Euclidean distance overlays a segmentation map over the image without having relevance to the actual content present in the image. Moreover, large diversity in the image (light conditions/high density portions) result in unavoidable distortion. Therefore, we hypothesize to use accurate distance measure which presents content relevant information of the visual scene. For this reason, we extend the functionality of SLIC by replacing the Euclidean distance measure with four potential similarity measures including chessboard, cosine, Minkowski, and geodesic and named it as SLIC+. These distance measures have been used in the literature integrated in clustering algorithms for synthetic textual data clustering where studies mentioned to render reasonable results for focused problem solving [44]. However, we use these similarity measures to investigate the effects on visual images using SLIC approach. Prior to implementation, a brief introductory discussion will help understand the overall integration and foundation for choosing these similarity measures. The distance measures are basically the distance transforms applied on different images, specifying the distance from each pixel to the desired pixel. For uniformity and easy understanding, let pixel $p_{1}$ and $p_{2}$ have the coordinates $\left(x_{1},\;y_{1}\right)$ and $(x_{2},\;y_{2}),$ respectively.

**Chessboard:** This measure calculates the maximum distance between vectors. This can be referred to measuring path between the pixels based on eight connected neighborhood whose edges are one unit apart. The chessboard distance along any co-ordinate is given by identifying maximum, as presented in Equation (4).
(4)$$D_{chess}=\max\left(\left|x_{2}-x_{1}\right|,\left|y_{2}-y_{1}\right|\right)$$*Rationale of Consideration*: Since the problem with existing similarity measures is loss of information, chessboard is one of the alternate to be incorporated in super-pixel creation base. This measure is considered as it takes into account information of eight connected neighbors of pixels under consideration. However, it might add computational overhead due to the same.**Cosine:** This measure calculates distance based on the angle between two vectors. The cosine angular dimensions counteract the problem of high dimensionality. The inner angular product between the vectors turns out to be one if vectors were previously normalized. Cosine distance is based on cosine similarity which is then plugged-in distance equation. Equations (5) and (6) shows calculation of cosine distance between pixels.
(5)$$cosine\;similarity=\frac{p_{1}.p_{2}}{\sqrt{p_{1}{}^{2}}\sqrt{p_{2}{}^{2}}}$$
(6)$$D_{cosine}=1-cosine\;similarity\;$$*Rationale of Consideration*: One of the aspects of content aware similarity measure is to retain the angular information thus we attempted to incorporate this measure. The resulting super-pixels are expected to retain the content relevant boundaries. However, this measure does not consider magnitude of the vectors/pixels due to which boundary performance might fall.**Minkowski:** This measure is a bit more intricate. It can be used for normed vector spaces, where distance is represented as vector having some length. The measure multiplies a positive weight value which changes the length whilst keeping its direction. Equation (7) presents distance formulation of Minkowski similarity measure.
(7)$$D_{min}={\left({\left|p_{2}-p_{1}\right|}^{µ}\right)}^{1/µ}$$Here µ is the weight, if its value is set to 1 the resultant measure corresponds to Manhattan distance measure. µ=2, refers to euclidean and µ=∞, refers to chessboard or Chebyshev distance measure.*Rationale of Consideration*: As user-control in respective application is desired, Minkowski similarity provides the functionality by replacing merely one parameter which changes the entire operationality without changing the core equations. However, here we still have problems relating to the retainment of angular information.**Geodesic:** This measure considers geometric movements along the pixel path in image space. This distance presents locally shortest path in the image plane. Geodesic distance computes distance between two pixels which results in surface segmentation with minimum distortion. Efficient numerical implementation of geodesic distance is achieved using first order approximation. For approximation parametric surfaces are considered with n number of points on the surface. Given an image mask, geodesic distance for image pixels can be calculated using Equation (8).
(8)$$D_{geo}=\underset{P_{x_{i},x_{j}}}{\min}\int_{0}^{1}D(P_{x_{i},x_{j}}\left(t\right)\|{\dot{P}}_{x_{i},x_{j}}\left(t\right)\|dt$$
where $P_{x_{i},x_{j}}\left(t\right)$ is connected path between pixel $x_{i},x_{j}$, provided *t* = 0,1. The density function $D\left(x\right)$ increments the distance and can be computed using Equation (9).
(9)$$D\left(x\right)=e^{\frac{E\left(x\right)}{\upsilon }},\;E\left(x\right)=\frac{\left\|▽I\right\|}{G_{\sigma }*\|▽I\|+\gamma^{\prime }}$$
where υ is scaling factor, $E\left(x\right)$ is edge measurement also provides normalization of gradient magnitude of image ‖▽I‖. $G_{\sigma }$ is the Gaussian function with its standard deviation being σ. γ minimizes the effect of weak intensity boundaries over density function. *D*(*x*) always produces constant distance, for homogeneous appearing regions if *E*(*x*) is zero *D*(*x*) becomes one.*Rationale of Consideration*: For shape analysis by computing distances geodesic has been the natural choice. However, computing geodesic distance is computationally expensive and is susceptible to noise [44]. Therefore, to overcome effect of noise geodesic distance should be used in amalgamation of Euclidean properties to retain maximum possible information in terms of minimum distance among pixels and their relevant angles.

The mentioned distance measures for identification of similarity among pixels based on pixel proximity provides different functionality features including extraction of information based on the 4-connected and 8-connected pixel neighborhood, and incorporation of geometric flows to keep track of angular movements of image pixels. However, none of these similarity measures provide balanced equation with integrated features of optimal boundary extraction based on connected neighbors and their angular movements. Thus, we hypothesize boundary extraction to be more accurate and intricate in presence of a similarity measure which provides greater information of spatial component provided by neighborhood pixels along with geometric flows.

### 3.5. SLIC++ Proposal

#### 3.5.1. Euclidean Geodesic—Content-Aware Similarity Measure

Considering the simplicity and fast computation as critical components for segmentation, the proposed algorithm uses fusion of Euclidean and geodesic distance measures. The depiction of Euclidean and geodesic similarity is presented in Figure 1, where straight line shows Euclidean similarity while curved line shows geodesic similarity. Since using only Euclidean similarity loses the context information due to usage of straight-line distance and geodesic similarity focuses more on the actual possible path along the pixels. We propose the fusion of both the similarities to extract accurate information of image pixels and their associations.

Using the same logic as SLIC, we propose a normalized similarity measure. The normalization is based on the interval S between the pixel cluster centers. To provide the control over super-pixel same variable m is also used. Beforehand the contribution of Euclidean and geodesic distance in final similarity measure cannot be determined in terms of optimized performance. Hence, we have introduced two weight parameters for proposal of final similarity measure which based on weighted combination of Euclidean and geodesic distance. The proposed similarity measure is presented in Equation (10).
(10)$$D_{ca}=w_{1}\left(d_{1}+\frac{m}{S}d_{2}\right)+w_{2}\left(d_{3}+\frac{m}{S}d_{4}\right)$$
where $D_{ca}$ is content-aware distance measure, $d_{1}$ and $d_{2}$ are same as $d_{s}$ and $d_{c}$ (Equations (5) and (6)) calculating the Euclidean distance for spatial and color component of image pixels. Variables $d_{3}$ and $d_{4}$ presents color and spatial component distance calculation using geodesic distance equation 8. Specifically, $d_{3}$ represents geodesic calculation of color components of image pixel and $d_{4}$ represents geodesic calculation of spatial components of pixel. Here, again we introduce similar normalization as of SLIC using variable S and m to provide control super-pixel compactness using geometric flows. The weights $w_{1}$ and $w_{2}$ further provides user control to choose the contribution of Euclidean and geodesic distance in final segmentation. These weights provide user flexibility, and these values can be changed based on the application. Moreover, these weights can be further tuned in future studies.

#### 3.5.2. Proposal of Content-Aware Feature Infusion in SLIC

The SLIC++ is proposed to extract the optimal information from a visual scene captured in semi-dark scenarios. Nevertheless, the same algorithm holds for any type of image if the objective is to retrieve maximum information from the image space. The steps involved in computing super-pixels are written in SLIC++ algorithm (refer Algorithm 1). Basically, super-pixels are perceptual cluster computed based on pixel proximity and color intensities. Some of the parameters include: K—being the number of super-pixels, N—total number of pixels, A—approximate number of pixels also called area of super-pixel, and S—length of super-pixel.
**Algorithm 1. SLIC++ Algorithm**1: Initialize K cluster center with seed (si)i=1K defined at regular intervals S2: Move cluster centers in n×n pixel neighborhood to lowest gradient3: ***Repeat***4:    For each cluster center si do5:     Assign the pixel from 2S×2S in a square window or CVT using distance measure given by Equation (3).6:     Assign the pixel from 2S×2S in a square window or CVT using distance measure given by Equation (10).7:   ***End for***8: Compute new cluster centers and residual error εrr (distance between previous centers and recomputed centers).9: Until εrr<= threshold10: Enforce connectivity.

Keeping simplicity and fast computation intact we present SLIC++ algorithm, here only one of the steps mentioned on step 5 or 6 will be used. If step 5 is implemented, i.e., distance measure given by Equation (3) is used entire functionality of SLIC algorithm is implemented. Whereas, if step 6 is implemented, i.e., distance measure given by Equation (10) is used entire functionality of SLIC algorithm is implemented.

Initialization and Termination

For initialization a grid of initial point is created separated by distance S in each direction as seen in Figure 2. The number of initial centers is given as parameter K. Placement of initial center in restricted squared grids can result in error if the initial center is placed on the edge of image content. This initial center is termed as confused center. To overcome this error gradient of the image is computed and the cluster center is moved in the direction of minimum gradient. The gradient is computed with 4-neighboring pixels and the centroid is moved. To solve this mathematically L2 Norm distance is computed among four connected neighbors of center pixel. The gradient calculation is given by Equation (11).
(11)$$G\left(x,y\right)=\|\left(x+1,y\right)-{\left(x-1,y\right)\|}^{2}+\|\left(x,y+1\right)-{\left(x,y-1\right)\|}^{2}$$

$G\left(x,y\right)$ is the gradient of center pixel under consideration.

The gradient of the image pixels is calculated until stability where pixels stop changing the clusters based on the initialized clusters. Overall, the termination and optimization is controlled by parameter ‘n’ which represents number of iterations the overall SLIC algorithm goes through before finally resulting in super-pixel creation of the image. To keep the uniformity in presented research we have selected ‘*n*’ as 10 which has been a common practice [11,14,29,32].


**How it works?**


The incoming image is converted to CIELAB space. The user provides information of all the initialization parameters including ‘K’, ‘m’, ‘n’. Referring to the algorithm steps presented in SLIC++ algorithm. Step 1, places K number of super-pixels provided by user on an equidistant grid. This grid is created separated by S, where S is given by NK, N is total number of image pixels. Step 2 performs reallocation of initial seed takes places subjected to the gradient condition to overcome the effect of initial centers placed over the edge pixels in image. Step 3 through step 7, are iteratively executed till the image pixels stop changing the clusters based on the cluster centers/seeds. Steps 5 or 6 are chosen based for respective implementation of SLIC or SLIC++ vice versa. Step 5 and 6 basically performs clustering over the image pixels based on different distance measures. If user opts for SLIC then Euclidean distance measure is used (base functionality). If user opts for SLIC++ then proposed hybrid distance measure is used. Step 8 checks if the new cluster center after every iteration of clustering is different than the previous center (distance between previous centers and recomputed centers). Step 9 keeps track of the threshold value for iterations as specified by the parameter ‘*n*’. Step 10 enforces connectivity among the created super-pixel/clusters of image pixels.

The simple difference in the implementation of SLIC and SLIC++ lies in the usage of distance measure being used for the computation of image super-pixels. The presented research shows merely changing the distance measure to content-aware computational distance measure leads to better accuracy of results against the ground-truth for semi-dark images.

b.Algorithm Complexity

The proposed algorithm follows the same steps as of Basic SLIC by introducing a new content-aware distance equation, thus the complexity of the proposed SLIC++ remains the same without any addition of new parameters, except the weights associated to the Euclidean and geodesic distance. These weights are merely scaler values to be taken into account in the core implementation of content-aware variant of SLIC, i.e., SLIC++. Hence, the complexity for the pixel manipulation is up to $O\left(N\right)$ where N is the total number of image pixels. With the minimum possible imposed requirements of computation SLIC++ manages to find accurate balance of implementation with infused functionality of Euclidean and geodesic distance. This fusion results in optimal boundary detection verified in terms of precision-recall in Section 4.

## 4. Validation of the Proposed Algorithm

### 4.1. Experimental Setup and Implementation Details

Following the proposed algorithm and details of implementation scheme, SLIC++ is implemented in MATLAB. The benchmarking analysis and experiments are conducted in MATLAB workspace version R2020a using the core computer vision and machine learning toolboxes. For experiments, the semi-dark images of Berkeley dataset have been used. The reported experiments are conducted on processor with specs core i7 10750H CPU, 16 GB RAM and 64-bit operating system.

The images are extracted form a folder using Fullfile method then incoming RGB images are converted in CIELAB space. After that parameter initialization takes place to get the algorithm started. Based on the number of K seed are initialized on the CIELAB image space and the condition relating to the gradient is checked using several different built-in methods. After that each pixel is processed using the proposed similarity measure and super-pixels are created until the threshold specified by user is reached. Similarly, the performance of reported state-of-the-arts is checked using the same environmental setup using the relevant parameters. Finally, the reported boundary performance is reported in form of precision recall measure to check the boundary adherence of super-pixel methods including Meanshift, SLIC and SLIC++. For analysis in terms of precision recall bfscore method is used which takes in the segmented image, ground-truth image and compares the extracted boundary with the ground-truth boundary by returning parameters precision, recall and score.

### 4.2. Parameter Selection

In this section we introduce the parameter associated with Meanshift, SLIC and SLIC++. Starting off with the proposed algorithm, SLIC++ uses several parameters as of Basic SLIC. Scaling factor m is set to 10, threshold on the iteration is set to value 10 represented by variable n and parameter S is computed based on N number of image pixels divided by user defined number of super-pixels in terms of variable K. The variable K provides user control for the number of super-pixels. Compact super-pixels are created as the value of K is increased but it increases the computational overhead. We have reported the performance using four different set of values of K, i.e., 500, 1000, 1500 and 2000. All these parameters including m, n and K are kept same as for the basic SLIC experiments. However, there are some additional parameters associated with SLIC++ which are $w_{1}$ and $w_{2}$ and their values are set to 0.3175 and 0.6825, respectively. The weights are cautiously picked based on trial-and-error experimentation procedure. The images were tested for a range of different weights. The weight values were varied to have weight ratios including 10:90, 30:70, 50:50, 70:30, and 90:10 for Euclidean and Geodesic distance, respectively. The ratio of 30:70 retains empirically maximum and perceptually meaningful super-pixels resulting in the optimal performance against the ground-truth. For Meanshift implementation the bandwidth parameter is set to 16 and 32, keeping rest of the implementation parameter default. Table 2 shows the averaged performance of the proposed SLIC++ algorithm acquired by varying different values of weights for random test cases.

Empirically optimized performance of SLIC++ over 30:70 weight ratio for Euclidean and Geodesic distance hybridization has been tabulated in Table 2 row number 4, 9, 14 (formatted bold and italics). Moreover, the parameter values have been set as *K* = 500, *m* = 10 and *n* = 10 for the conducted experiments.

### 4.3. Performance Analysis

For performance analysis we considered two different experimental setups including qualitative analysis and quantitative analysis. Initially, we extended and analyzed the performance of SLIC with different distance measures to propose the most relevant distance measure for optimal boundary extension in semi-dark images. Then we compare the proposed algorithm with state-of-the-art super-pixel segmentation algorithms. The detail of the analysis is presented in following sub-sections.

#### 4.3.1. Numeric Analysis of SLIC Extension with Different Distance Measures

For the detailed analysis of the proposed algorithm, we first compare the performance of basic SLIC with the variants of SLIC+ proposed in this study. The evaluation is presented in form of precision recall. For the optimal boundary detection greater values of precision are required. High precision rates relate to low number of false positives eventually resulting in high chance of accurate boundary retrieval, whereas high recall rates are relevant to matching of ground-truth boundaries to segmented boundary. Mathematically, precision is probability of valid results and recall is probability of detected ground-truths data [42]. For analysis of image segmentation modules, both high precision and recall are required to ensure maximum information retrieval [45].

Table 3 shows performance analysis of basic SLIC and its variants over randomly picked semi-dark images.

Table 2 depicts all the extension of SLIC perform better in terms of precision-recall. The parameters are kept uniform for all the experiments specifically parameter m and n as in SLIC [11]. Moreover, there is up to 3–9% gain in precision percentage using SLIC++ as compared to the basic SLIC algorithm. The relevant scores based on precision and recall also shoot up by margin of 5–9% using SLIC++ (row 1 vs. 6 and row 7 vs. 12). However, the performance of other variants of SLIC is subjective to dimensions of incoming data, magnitudes, and memory overload. There usually is no defined consensus regarding best generalized performer in terms of similarity measure so far [44]. Thus, we propose an integration of two similarity measures which takes into account minimal processing resources and still provides optimal boundary detection.

For further detailed qualitative analysis using the same test cases by changing the number of super-pixels we extend the analysis of SLIC versus SLIC++. The precision recall and score graphs are shown in Figure 3.

In Figure 3, solid lines represents performance of SLIC and SLIC++ for Test case 1 and dashed lines represents performance for Test case 2. Figure 3a shows precision curves of SLIC++ are substantially better than the SLIC presented by brown (dark and light) lines for test case 1 and 2, respectively. Figure 3b shows the SLIC++ recall is less the resulting recall of SLIC for the same images. Subsequently, based on the precision, recall and the final scores SLIC++ outperforms basic SLIC on semi-dark images. For number of pixels set to 1000 there is a drop observed in precision and recall of SLIC++, this behavior can be attributed to accuracy measure’s intolerance, i.e., even mutual refinements may result in low precision and recall values [45]. Nevertheless, performance for retrieval increases with increasing number of super-pixels and SLIC++ outperforms SLIC up to margin of 10%.

#### 4.3.2. Comparative Analysis with State-of-the-Art

For the benchmarking of SLIC++ two different algorithms, i.e., SLIC and Meanshift are considered. To investigate the performance of SLIC and SLIC++ for the analysis over entire Berkeley dataset (semi-dark images), we set the number of super-pixels to 1500. The number of super-pixels is chosen 1500 because the peak performance of both the algorithms in experiment for test case 1 and 2 (refer Figure 3) is achieved by setting this parameter to value 1500. For the comparative analysis we also used Meanshift algorithm with input parameter, i.e., bandwidth set to 32. The bandwidth of Meanshift decides the complexity of the algorithm as this value is decreased the segmentation becomes more intricate with the overhead of computational complexity. To maintain computational resources throughout the experiment and keeping it uniform the parameters are chosen. The summary statistics of the obtained super-pixel segmentation results are shown in Table 4. The numerals presented in table are averaged values of precision, recall, and scores obtained for 316 images separately. The average precision, recall, and scores are presented in Table 4.

Table 4 shows SLIC++ achieves average percentage score up to 54%, whereas SLIC maintains a score of 47%. Finally, Meanshift achieves a score of 55%, which is greater than SLIC++ but as stated earlier for segmentation application greater values of precision and recall are required. So, comparing the recall of SLIC++ versus Meanshift a huge difference is observed. This difference is in terms of low recall of Meanshift which means algorithm fails to capture salient image structure [45] which is not desired for semi-dark image segmentation.

#### 4.3.3. Boundary Precision Visualization against Ground-Truth

To validate the point-of-view relating to high precision and high recall we present perceptual results of Meanshift, SLIC and SLIC++. Notice that, the high precision means the algorithm has retrieved most of the boundary as presented by the ground-truth, whereas high recall means most of the salient structural information is retrieved from the visual scene. Meanshift resulted in a minimum recall, which hypothetically means the structural information was lost. Table 5 presents how Meanshift, SLIC, and SLIC++ performed in terms of perceptual results for visual information retrieval. The reported results are for parameters *K* = 1500 for SLIC and SLIC++ and bandwidth = 32 for Meanshift.

As super-pixels are not just about the boundary detection, resulting applications also expect the structural information present in the visual scene. Consequently, we are not just interested in the object boundaries but also the small structural information present in the visual image specifically semi-dark images. Table 5 shows SLIC and SLIC++ not only retrieves boundaries correctly with minimal computational power consumed but also retrieves the structural information. Column 4 shows the fact by mapping prediction over ground-truth image. For test case 1, in column 4 row id 3 Meanshift fails to extract the structural information as few green lines are observed. Whereas, for the same image SLIC and SLIC++ perform better as a lot of green textured lines are observed (refer column 4 row id 1 and 2). Meanwhile for test case 2, all three algorithms perform equally likely. Similar performance is observed with test case 3, SLIC and SLIC++ retains structural information better than Meanshift. Since Meanshift resulted in minimum recall over the entire semi-dark Berkeley dataset (refer Table 4) it does not qualify to be a good fit for super-pixel segmentation. The reason is less reliability of structural information fetching and its performance is highly subjective to the incoming input images.

#### 4.3.4. Visualizing Super-Pixels on Images

For one more layer of subjective analysis of super-pixel performance we present super-pixel masks in this section. Initially, we present the input image in Figure 4 with the highlighted boxes to look closely for retrieval of structural information from the image. Here, the red box shows the texture information present on the hill whereas the green box shows water flowing in a very dark region of the semi-dark image.

Using the input image presented in Figure 4, we conducted experiments by changing the initialization parameters of all three algorithms. Table 5 shows the perceptual analysis visualizing the retrieval of salient structural information.

Table 6 shows that Meanshift extracts the boundaries correctly, whereas it loses all the contextual information when the bandwidth parameter is set to 32. This loss of information is attributed to low recall scores, whereas decreasing the value of the bandwidth increases the computational complexity and at the cost of additional complexity Meanshift now retrieves contextual information. SLIC and SLIC++ with minimal computational power retains structural information as seen in the red and green boxes in rows 1 and 2 of Table 5. Moreover, as the number of super-pixels ‘*K*’ increases, better and greater structural information retrieval is observed.

Figure 5 shows a zoomed in view of the super-pixels created by SLIC and SLIC++, residing in the red box. Here, we can see that SLIC++ retrieves content-aware information and SLIC ends up creating circular super-pixels (Figure 5a) due to the content irrelevant distance measure being used in its operational discourse.

#### Key Takeaways

The benchmarking analysis shows that the proposed algorithm SLIC++ achieves robust performance over different cases. The results of SLIC++ are more predictable as compared to the state-of-the-art methods Meanshift and SLIC. The performance of Meanshift is highly subjective as the recall keeps changing. Less recall values eventually result in less scores at the cost of information loss. Whereas, SLIC achieves 7% less scores and 8% less precision values in terms of boundary retrieval. The results of SLIC++ indicate that the proposed content-aware distance measure integrated in base SLIC demonstrates superior results. The significant improvement to the existing knowledge of super-pixel creation research is hybridization of proximity measures. Based on the comprehensive research it is seen that the hybrid measure performs better than the singular proximity measure counterparts of the same algorithm. These measures substantially control the end results of super-pixel segmentation in terms of accuracy. The proposed hybrid proximity measure carefully finds a balance between the two existing distance measure and performs clustering over image pixels making sure to retain content-aware information.

## 5. Limitations of Content-Aware SLIC++ Super-Pixels

The super-pixel segmentation algorithms are considered pre-processing step for wide range of computer vision applications. To obtain the optimal performance of sophisticated applications, the base super-pixel algorithm SLIC uses set of input parameters. These parameters allow the user control over different aspects of image segmentation. The idea is to extract uniform super-pixels throughout the image grid to maintain reliable learning statistics throughout the process. To make this possible the SLIC initially allows user to choose number of pixels ‘*K*’ (values ranging from 500–2000), parameter ‘*m*’ (where *m* = 10) which decides the extent of enforcement of color similarity over spatial similarity, number of iterations ‘N’ (where N = 10) which decides the convergence of the algorithm, neighborhood window ‘*w*’ (where *w* = 3) for gradient calculation to relocate cluster centers (if placed on edge pixel). This makes four input parameters for the base SLIC, whereas the proposed extension SLIC++ introduces two more weight parameters, $w_{1}$ and $w_{2}$ (0.3175 and 0.6825, respectively), to decide the impact of each distance measure in the hybrid distance measure. All these parameters significantly control the accuracy of segmentation results. Incorrect selection of these parameters leads to overall poor performance. Hence, for diverse applications, initial parameter search is necessary, which in turn requires several runs. For the reported research, using the state-of-the-art segmentation dataset, i.e., Berkeley dataset we chose the parameters as selected by the base SLIC. These parameters offer good performance over the image size of 321 × 481 or 481 × 321, whereas, as we increased the resolution of images during the extended research we observed that a higher value of ‘*K*’ is required for better segmentation accuracy.

For the existing research, we conducted experiments focused to identify the gains associated with usage of the proposed content-aware distance measure over the straight line distance measure. For the extended research, the input parameters shall be considered for optimization.

## 6. Emerging Successes and Practical Implications

Several decades of research in computer vision for boosted implementations resulting in fast accurate decisions, super-pixels have been a topic of research for long time now. The super-pixel segmentation is taken as entry stage providing pre-processing functionality for sophisticated intelligent workflows such as semantic image segmentation. To speed up the overall process of training and testing of these intelligent systems super-pixels are probable to provide remedies. As the intelligent automated vision systems have critical applications in medicine [46,47], manufacturing [48], surveillance [49], tracking [2] and so on. For this reason, fast and accurate visual decision are required. As the environmental conditions in form of visual dynamicity is challenging task to tackle by pre-processing modules. These modules are required to provide reliable visual results. Many super-pixel creation algorithms have been proposed over time to solve focused problems of image content sparsity [30], initialization optimization [28], and accurate edge detection [38]. However, the topic of the lightning condition in this domain remains untouched and needs attention. The dynamic lightning condition is a key component in autonomous vehicles, autonomous robots, surgical robots. The Berkeley dataset is comprised of images of different objects, ranging from humans, flowers, mountains, animals and so on. The conducted research holds for applications of autonomous robots and autonomous vehicles. However, the proposed algorithm is backed by the core concepts of image segmentation. For this reason, the presented work can be extended for variety of applications. Depending on the nature of application, the ranges of input parameters would be changed based on the required sensitivity of the end results, such as for the segmentation application in the medical domain compact where content-aware information is required. Consequently, the input values including the number of super-pixels to be created will be carefully selected. To handle the pre-processing problems associated with dynamic lightning conditions focusing autonomous robotics, the proposed extension of SLIC is a good fit. SLIC++ imposing minimum prerequisite conditions provides direct control over the working functionality and still manages to provide optimal information retrieval from the visual scenes for not only normal images but rather inclusive of semi-dark images.

## 7. Conclusions and Future Work

### 7.1. Conclusions

In this paper, we introduced a content-aware similarity measure which not only solved the problem of boundary retrieval in semi-dark images but is also applicable to other image types such as bright and dark. The content-aware measure is integrated in SLIC to create content-aware super-pixels which can then be used by other automated applications for fast implementations of high-level vision task. We also report results of integration of SLIC with existing similarity measures and describe their limitations of applicability for visual image data. To validate out proposed algorithm along with the proposed similarity measure, we conduct qualitative and quantitative evaluations on semi-dark images extracted from Berkeley dataset. We also compare SLIC++ with state-of-the-art super-pixel algorithms. Our comparisons show that the SLIC++ outperforms the existing super-pixel algorithms in terms of precision and score values by a margin of 8% and 7%, respectively. Perceptual experimental results also confirm that the proposed extension of SLIC, i.e., SLIC++ backed by content-aware distance measure outperforms the existing super-pixel creation methods. Moreover, SLIC++ results in consistent and reliable performance for different test image cases characterizing a generic workflow for super-pixel creation.

### 7.2. Future Work

For the extended research, density-based similarity measures integrated with content-aware nature may lead the future analysis. The density-based feature is expected to aid the overall all working functionality with noise resistant properties against the noisy incoming image data. Moreover, another aspect is the creation of accurate super-pixels in the presence of non-linearly separable data properties. Finally, the input parameter selection for the initialization of SLIC variants depending on the application domain and incoming image size remains an open area of research.

## Figures and Tables

**Figure 1 sensors-22-00906-f001:**
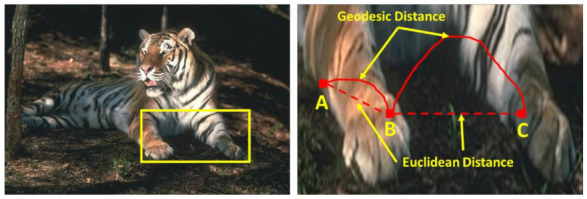
Irrelevance of Euclidean distance measure for super-pixel creation relating to image content.

**Figure 2 sensors-22-00906-f002:**
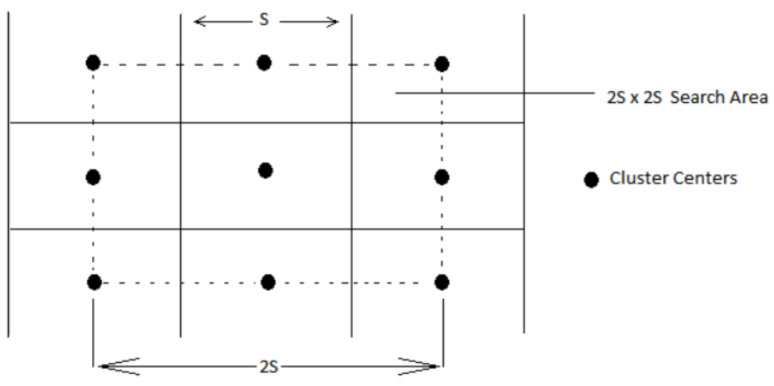
Restricted Image search area for super-pixel creation specified by input argument for image window under consideration [31].

**Figure 3 sensors-22-00906-f003:**
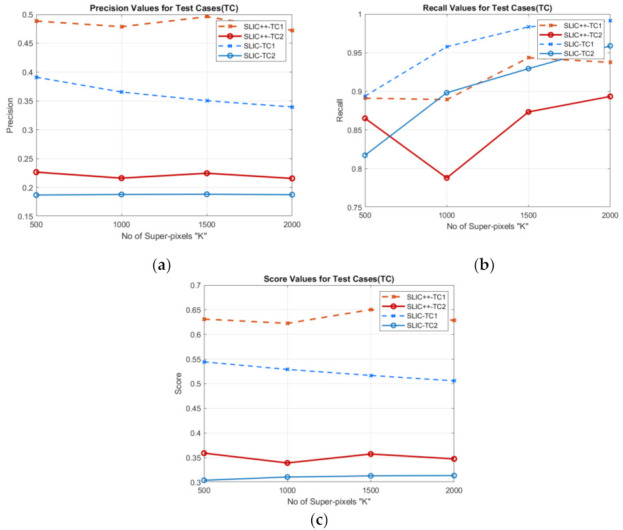
SLIC v/s SLIC++ performance over different number of pixels: (**a**) precision values; (**b**) recall value; (**c**) score values.

**Figure 4 sensors-22-00906-f004:**
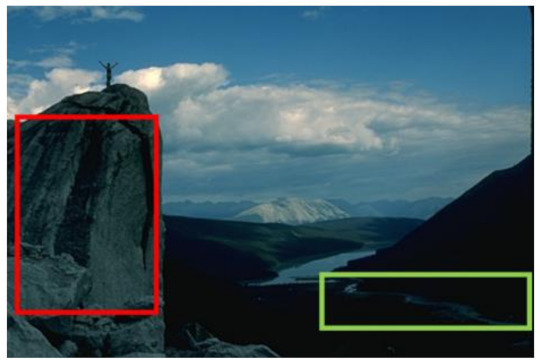
Input image with highlighted regions for detailed analysis.

**Figure 5 sensors-22-00906-f005:**
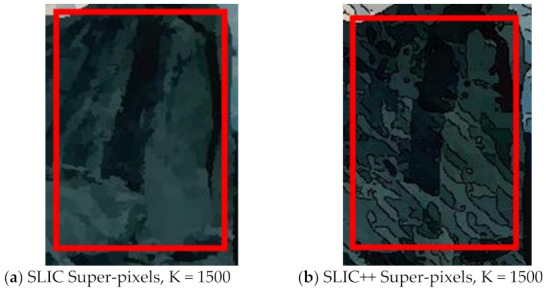
Zoomed in view of test case image for content-aware super-pixel analysis created by SLIC++ (**b**) against SLIC (**a**).

**Table 1 sensors-22-00906-t001:** Critical analysis of gradient-ascent super-pixel creation algorithms.

Method	Complexity	Pixel Manipulation Strategy	Distance Measure	Dataset	Semi-Dark Image Mentions	Year	Ref.
Meanshift	$O\left(N^{2}\right)$	Mode seeking to locate local maxima	Euclidean	Not mentioned	✘	2002	[17]
Medoidshift	$O\left(N^{2}\right)$	Approximates local gradient using weighted estimates of medoids.	Euclidean	Not mentioned	✘	2007	[18]
Quickshift	$O\left(dN^{2}\right)$	Parzen’s density estimation for pixel values	Non-Euclidean	Caltech-4	✘	2008	[19]
TurboPixel	$O\left(N\right)$	Geometric flows for limited pixels	Gradient calculation for boundary pixels only	Berkeley Dataset	✘	2009	[20]
Scene Shape Super-pixel (SSP)	ON32 logN	Shortest path manipulation with prior information of boundary.	Probabilistic modeling plus Euclidean space manipulation	Dynamic road scenes. No explicit mentions of semi-dark images but we suspect presence of semi	✓	2009	[21]
Compact Super-pixels (CS)	-	Approximation of distance between pixels and further optimization with graph cut methods	Euclidean	3D images	✘	2010	[22]
Compact Intensity Super-pixels	-	Same as CS, With added color constant information.	Euclidean	3D images	✘	2010	[22]
SLIC	$O\left(N\right)$	Gradient optimization after every iteration	Euclidean	Berkeley Dataset	✘	2012	[11]
SEEDS	-	Energy optimization for super-pixel is based on hill-climbing.	Euclidean	Berkeley Dataset	✘	2012	[23]
Structure Sensitive Super-pixels	$O\left(N\right)$	Super-pixel densities are checked, and energy minimization is conformed.	Geodesic	Berkeley Dataset	✘	2013	[16]
Depth-adaptive super-pixels	$O\left(N\right)$	Super-pixel density identification, followed by sampling and finally k-means to create final clusters	Euclidean	RGB-D dataset consisting of 11 images	✘	2013	[24]
Contour Relaxed Super-pixels	$O\left(N\right)$	Uses pre-segmentation technique to create homogeneity constraint	Not mentioned	Not mentioned	✘	2013	[25]
Saliency-based super-pixel	$O\left(N\right)$	Super-pixel creation followed by merging operator based on saliency.	Euclidean	Not mentioned	✘	2014	[26]
Linear Spectral Clustering	$O\left(N\right)$	Two-fold pixel manipulation strategy of optimization based on graph and clustering based algorithms.	Euclidean	Berkeley Dataset	✘	2015	[27]
Manifold SLIC	$O\left(N\right)$	Same as SLIC but with mapping over manifold.	Euclidean	Berkeley Dataset (Random)	✘	2016	[15]
BASS (Boundary-Aware Super-pixel Segmentation)	$O\left(NlogN\right)$	Extension of SLIC initially creates boundary then uses SLIC with different distance measures, along with optimization of initialization parameters.	Euclidean + Geodesic	Fashionista, Berkeley Segmentation Dataset (BSD), HorseSeg, DogSeg, MSRA Salient Object Database, Complex Scene Saliency Dataset (CSSD) and Extended CSSD (ECSSD)	**✓**	2016	[28]
BSLIC	$O\left(6N\right)$	Extension of SLIC initializes seed within a hexagonal space rather than square	Euclidean	Berkeley Dataset	✘	2017	[29]
Intrinsic Manifold SLIC	$O\left(N\right)$	Extension of Manifold SLIC with geodesic distance measure	Geodesic	Berkeley Dataset (Random)	✘	2017	[30]
Similarity Ratio based Super-pixels	$O\left(N\right)$	Extension of SLIC. Proposes automatic scaling of coordinate axes.	Mahanlanobis	SAR Image dataset	**✓**	2017	[31]
Scalable SLIC	$O\left(N\right)$	Optimized initialization parameters such as ‘n’ number of super-pixels, focused research to parallelization of sequential implementation.	Euclidean	cyrosection Visible Human Male dataset	✘	2018	[32]
Content adaptive super-pixel segmentation	$O\left(N^{2}\right)$	Work on prior transformation of image with highlighted edges created by edge filters	Euclidean (with graph-based transformation)	Berkeley Dataset	✘	2019	[33]
BASS (Bayesian Adaptive Super-Pixel Segmentation)	$O\left(N\right)$	Uses probabilistic methods to intelligently initialize the super-pixel seeds.	Euclidean	Berkeley Dataset	✘	2019	[34]
Super-pixel segmentation with fully convolutional networks	$O\left(N^{No.\;of\;layers}\right)$	Attempts to use neural networks for automatic seed initialization over grid.	Euclidean	Berkeley Dataset, SceneFlow Dataset	✘	2020	[35]
Texture-aware and structure preserving Super-pixels	$O\left(N\right)$	The seed initialization takes place in circular grid.	Three different distance measure (without explicit details)	Berkeley Dataset	✘	2021	[36]
Efficient Image-Warping Framework for Content-Adaptive Super-pixels Generation	$O\left(N\right)$	Warping transform is used along with SLIC for creation of adaptive super-pixels.	Euclidean	Berkeley Dataset	✘	2021	[37]
Edge aware super-pixel segmentation with unsupervised CNN	$O\left(N^{No.\;of\;layers}\right)$	Edges are detected using unsupervised convolutional neural networks then passed to super-pixel segmentation algorithms	Entropy based clustering	Berkeley Dataset	✘	2021	[38]

**Table 2 sensors-22-00906-t002:** Summary statistics of average performance of SLIC++ for varying weights.

Row	Ratio	$w_{1}$ (Euclidean)	$w_{2}$ (Geodesic)	Precision	Recall	Score
	Test Case 1 (Image ID = 14037):					
1	10:90	0.1123	0.8877	0.47882	0.88930	0.62248
2	70:30	0.6825	0.3175	0.38850	0.92210	0.54660
3	50:50	0.4863	0.5137	0.37780	0.93040	0.53740
** *4* **	** *30:70* **	** *0.3175* **	** *0.6825* **	** *0.48854* **	** *0.89124* **	** *0.63113* **
5	90:10	0.8877	0.1123	0.38840	0.87340	0.53770
	Test Case 2 (Image ID = 26031):					
6	10:90	0.1123	0.8877	0.21623	0.78808	0.33935
7	70:30	0.6825	0.3175	0.18370	0.82790	0.30070
8	50:50	0.4863	0.5137	0.18910	0.85000	0.31000
** *9* **	** *30:70* **	** *0.3175* **	** *0.6825* **	** *0.22661* **	** *0.86520* **	** *0.35920* **
10	90:10	0.8877	0.1123	0.18650	0.79000	0.30220
	Test Case 3 (Image ID = 108082):					
11	10:90	0.1123	0.8877	0.27023	0.89832	0.41548
12	70:30	0.6825	0.3175	0.21840	0.82160	0.34510
13	50:50	0.4863	0.5137	0.22640	0.86800	0.35920
** *14* **	** *30:70* **	** *0.3175* **	** *0.6825* **	** *0.28547* **	** *0.91629* **	** *0.43532* **
15	90:10	0.8877	0.1123	0.22360	0.79470	0.34900

**Table 3 sensors-22-00906-t003:** Performance analysis of SLIC extensions.

Row	*K*	*m*	*n*	Parameters		Score	Precision	Recall	Distance Measure
Test Case 1 (Image ID = 14037):									
1	500	10	10			0.54430	0.39120	0.89390	Euclidean—SLIC
2	500	10	10			0.61234	0.46563	0.89406	Chessboard—SLIC+
3	500	10	10			0.59713	0.44407	0.91118	Cosine—SLIC+
4	500	10	10	µ=4		0.62792	0.47345	0.93199	Min4—SLIC+
5	500	10	10			0.56128	0.43777	0.78186	Geodesic—SLIC+
**6**	**500**	**10**	**10**	$w_{1}=0.3175$	$w_{2}=0.6825$	**0.63113**	**0.48854**	**0.89124**	**Euclidean Geodesic—SLIC++**
Test Case 2 (Image ID = 26031):									
7	500	10	10			0.30420	0.18690	0.81740	Euclidean—SLIC
8	500	10	10			0.35454	0.22098	0.89623	Chessboard—SLIC+
9	500	10	10			0.35698	0.22329	0.88957	Cosine—SLIC+
10	500	10	10	µ=4		0.34057	0.20959	0.90798	Min4—SLIC+
11	500	10	10			0.33715	0.21369	0.79842	Geodesic—SLIC+
**12**	**500**	**10**	**10**	$w_{1}=0.3175$	$w_{2}=0.6825$	**0.3592**	**0.22661**	**0.86525**	**Euclidean Geodesic—SLIC++**
Test Case 3 (Image ID = 108082):									
13	500	10	10			0.35410	0.22720	0.80260	Euclidean—SLIC
14	500	10	10			0.42099	0.27720	0.87476	Chessboard—SLIC+
15	500	10	10			0.38368	0.24251	0.91811	Cosine—SLIC+
16	500	10	10	µ=4		0.42465	0.27694	0.91004	Min4—SLIC+
17	500	10	10			0.40382	0.26764	0.82216	Geodesic—SLIC+
**18**	**500**	**10**	**10**	$w_{1}=0.3175$	$w_{2}=0.6825$	**0.43532**	**0.28547**	**0.91629**	**Euclidean Geodesic—SLIC++**

**Table 4 sensors-22-00906-t004:** Summary statistics of average performance for Berkeley dataset.

Algorithm	Score	Precision	Recall
**SLIC**	0.47020	0.31604	0.97719
**SLIC++**	**0.54799**	**0.39776**	**0.93470**
**Meanshift-32**	0.55705	0.57573	0.68416

**Table 5 sensors-22-00906-t005:** Semi-dark perceptual results conforming boundary retrieval.

Row ID	Image	Groundtruth Image	Prediction	Prediction Map Compared with Groundtruth
**Test Case 1:**				
**1**	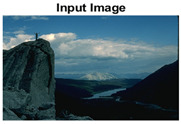	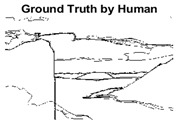	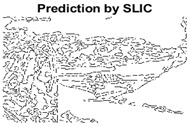	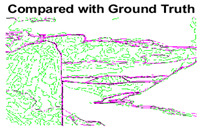
**2**	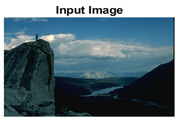	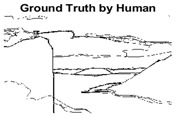	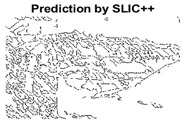	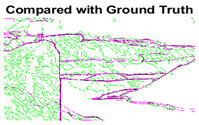
**3**	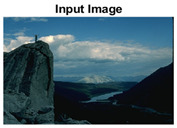	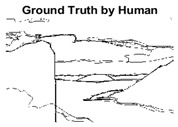	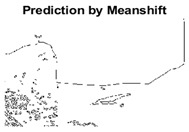	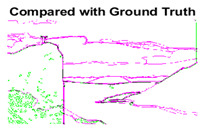
**Test Case 2:**				
**4**	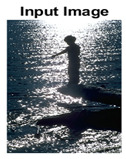	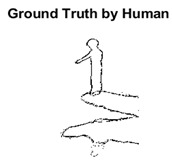	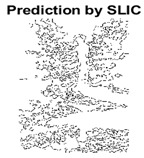	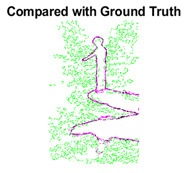
**5**	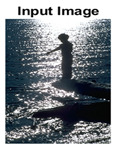	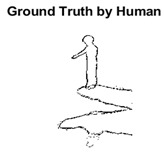	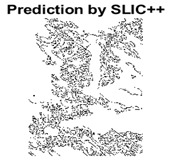	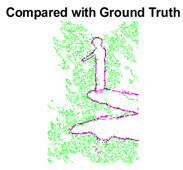
**6**	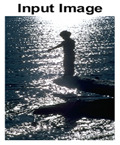	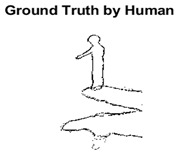	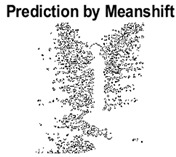	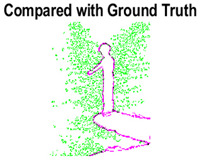
**Test Case 3:**				
**7**	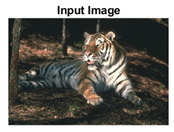	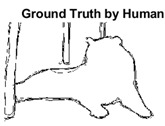	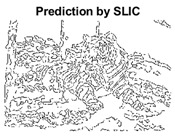	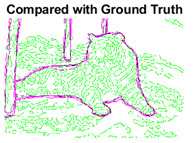
**8**	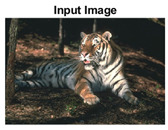	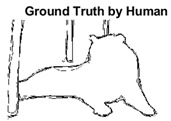	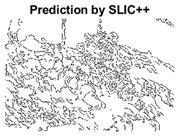	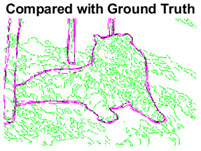
**9**	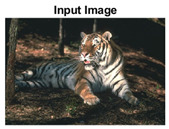	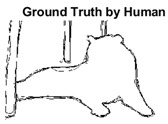	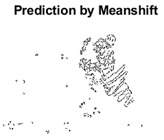	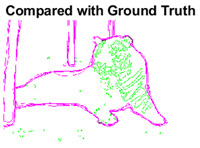

**Table 6 sensors-22-00906-t006:** Detailed perceptual analysis with increasing parameters.

Number of Super-Pixels/Algorithm	500	1000	1500	2000
**SLIC**	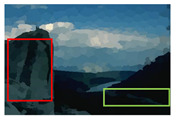	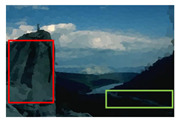	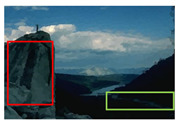	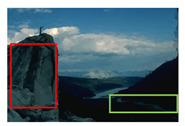
**SLIC++**	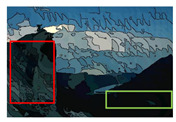	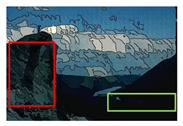	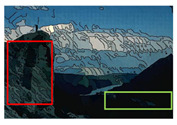	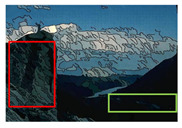
**Bandwidth/** **Meanshift**	**16**		**32**	
	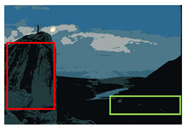		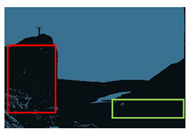	

## Data Availability

Following dataset is used. Berkeley: https://www2.eecs.berkeley.edu/Research/Projects/CS/vision/bsds/ (accessed on 27 September 2021).

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
