# Peer review of "Content-Aware SLIC Super-Pixels for Semi-Dark Images (SLIC++)"

_sensors, 2022, doi:10.3390/s22030906_

Round 1

Reviewer 1 Report

The authors should present their ideas and algorithm in a more compact and straightforward way. After going through this paper, I am still not very sure about the key improvement their extension in SLIC++, and in which application this improvement would help. Semantic segmentation is sometime a very application-dependent task. To which extend the detail i.e. the contextual details should be resolved is an open issue.  

Moreover, the SLIC as well as SLIC++ are not explained very clear. At least, after reading your paper, I cannot sure which cost function is optimized. 

Author Response

Dear Reviewer,

Thank you for your constructive reviews and suggestions. We believe that the updated manuscript addresses all the comments and can be now considered for publication.

Thanking you on behalf of all the authors.

Mehak

Reviewer 2 Report

  1. Line 15 – “it simplistic approach” – should be its simplistic approach, Line 18 – “in lower” – should be is lower, Line “efficient hold” – do the authors mean to say hold efficiency? Line 133: “The following paper is organized” – this should be written as the remainder of the paper is organized, Line 457: Eq 10 is off-center and “Eq. 10” encompasses 2 separate lines, Line574: there should be ‘of’ before “precision-recall”, Line 480 – do the authors mean to say “step 5 or step 6”? There are several such errors all throughout the manuscript. Authors are urged to take note of such sentence construction, typographical, grammatical, and spelling errors and correct them.
  2. The nomenclature SLIC++ is confusing – Why is it termed SLIC++ and not SLIC+?
  3. Line 30: “precision accuracy of 39.7%” - this is not understood – is precision 39.7% or accuracy 39.7%? Precision and accuracy are two separate metrics. Furthermore, a 39.7% accuracy seems rather on the lower end - how is the proposed method justifiable, under such low accuracy regimes?
  4. The manuscript initially focuses and spends a great deal discussing existing literature on “singular pixel manipulation strategies” – considering that the primary theme of the manuscript is to extend SLIC to SLIC++, this seems like an unnecessary addendum and can be shortened or briefed to the point where only literature relevant to the core topic of discussion should be kept and should be presented in crisp, textual form.
  5. Figure captions should be descriptive – this is currently lacking in the manuscript.
  6. Table 2 shows the performance of SLIC++ on 2 test images – how exactly are the values for the weights w_1 and w_2 obtained? Are these obtained through trial and error/empirically?
  7. There are only 2 test cases discussed – this seems a small number to draw generalized conclusions. This should be increased to incorporate sufficient variability in showcasing the performances of the different methods.
  8. Limitations of the proposed method are not explicitly discussed.

Author Response

(The authors gave the same response as above.)

Round 2

Reviewer 1 Report

I have no further comments.

Author Response

Respected Reviewer, 

We have carefully addressed your reviews. We hope you would be satisfied after this revision.

Thank you.

Reviewer 2 Report

While the authors have addressed most of my comments, the authors should address the following comments in greater detail:

  1. Line 560: “The ratio of 30:70 retains empirically maximum and perceptually meaningful super-pixels resulting the optimal performance against the ground-truth” – This should either be accompanied by a figure or a table that shows what the authors mean to say here.
  2. Line 560: “The ratio of 30:70 retains empirically maximum and perceptually meaningful super-pixels resulting the optimal performance against the ground-truth” – this should be - resulting in the optimal performance i.e., ‘in’ is missing. Line 711: “Hence, for diverse applications initially parameter search is required which in turn requires several runs.” – this should be – Hence, for diverse applications, initial parameter search is necessary, which in turn requires several runs. Line 715: “resolution of the images higher number of ‘K’” – this should be – resolution of images, a higher value of ‘K’. Line 774 – “image size remains open are of research.” – this should be – image size remains an open area of research. Authors are again urged to correct such sentence construction, grammatical and typographical errors in the revised manuscript as otherwise the presence of such errors leads to a severely degraded manuscript quality.
  3. As a response to my initial comment 7, the authors have added another single image as Test Case 3 – however, this is not what was expected. What was expected was either summary statistics of the entire validation dataset (which seems to be in Figure 3 – please confirm this) or an explicit mention in the text and/or in the Table caption how representative these test cases are and what’s the rationale behind choosing them as representative cases.

Author Response

Respected reviewer,

We have carefully addressed your suggestions. We hope you would be satisfied after this revision.

Thanks
